# Pediatric Health Access and Private Medical Insurance: Based on the Ecology of Medical Care in Korea

**DOI:** 10.3390/children9081101

**Published:** 2022-07-22

**Authors:** Dong-Hee Ryu, Yong-jun Choi, Jeehye Lee

**Affiliations:** 1Department of Preventive Medicine, Daegu Catholic University School of Medicine, Daegu 42472, Korea; ryudh@cu.ac.kr; 2Department of Social and Preventive Medicine, College of Medicine, Hallym University, Chuncheon 24252, Korea; ychoi@hallym.ac.kr; 3Institute of Social Medicine, College of Medicine, Hallym University, Chuncheon 24252, Korea; 4Department of Preventive Medicine, Eulji University School of Medicine, Daejeon 34824, Korea

**Keywords:** healthcare, insurance coverage, national health insurance

## Abstract

This study aimed to investigate pediatric health access by describing the ecology of medical care for children and adolescents in a medical environment where a well-balanced system between national health insurance (NHI) and private medical insurance (PMI) is required. Data from 2746 individuals aged 18 years old and younger were used. Of the participants, 87.3% had private medical insurance. Of the 1000 children, in an average month, 404 visited a clinic, 67 visited a hospital outpatient department (OPD), 49 visited an OPD in a tertiary hospital, 11 received emergency care, 5 received inpatient care in a hospital, and 9 were hospitalized. The generalized estimating equation models adjusted for age, sex, economic status, and pediatric comorbidity index were used for multivariate analysis. Receiving ambulatory care services in clinics was significantly more likely among children and adolescents with private medical insurance (adjusted odds ratio [aOR] = 1.16 [95% confidence interval [CI]: 1.00–1.35]). Receiving ambulatory care services in clinics was significantly more likely among indemnity type policyholders (aOR = 1.23 [1.05–1.45]) and single policyholders (aOR = 1.18 [1.00–1.37]). Countries with national health insurance schemes should continuously practice the proper regulation and management of PMI, including reviewing PMI compensation measures, NHI reimbursement standards, and consumers’ perspectives on NHI and PMI.

## 1. Introduction

Universal health coverage (UHC) allows every person to have access to necessary health services, from health promotion and prevention to disease treatment, rehabilitation, and palliative care, with the least financial difficulties [1]. Many countries focus on the establishment of UHC for their citizens by creating well-controlled government-sponsored health coverage. Even in the United States, where a private health insurance market is active, progress has been made by requiring private insurance companies to offer qualified health plans (QHPs) based on the requirements of the Affordable Care Act (ACA) [2,3].

All citizens in Korea have been insured under the National Health Insurance (NHI) system since 1989. According to the Organization for Economic Cooperation and Development (OECD) data, the 2018 health expenditure per capita in Korea was USD 3091.8 PPP and the percentage of gross domestic product (GDP) was 7.5% [4]. The 2018 health expenditure per capita in Germany was USD 6291.0 PPP and the percentage of GDP was 11.5%, while that of the United States was USD 10,528.50 PPP and 16.7%, respectively [4]. The share of government and compulsory schemes was 60.1% (USD 1857.1 PPP) in Korea, 84.5% (USD 5313.8 PPP) in Germany, and 82.8% (USD 8714.8 PPP) in the United States [4]. Although the contractual premise is guaranteed in Korea, the above figures indicate the necessity of examining what the Korean UHC system is pursuing regarding users’ financial burden and the appropriate level of insurance premise.

Many Koreans buy private medical insurance (PMI) products to alleviate the burden of out-of-pocket expenses, including co-payments for covered services. The various types of PMI products―fixed amount, indemnity, and mixed type―are being sold and each product additionally provides numerous types of special elements within the insurance contract. Moreover, consumers can buy as many PMI products as they want since there is no systematic restrictions regarding the purchase. According to a previous report, approximately 78% of Koreans were PMI insured in 2015, and the enrollment rate for indemnity type PMI was 64.9% [5]. For countries with government-sponsored health coverage, the size and role of the PMI market vary. In Germany, people buy PMI products if their income exceeds the compulsory insurance threshold [6]. The PMI market in the UK, where the National Health Service (NHS) system is provided, is also different from Korea. Many people buy PMI products to receive services not provided by the NHS or to use medical services immediately [5].

PMI has been widespread among Korean children and adolescents since the early 2000s when the demand for so-called “prenatal child insurance” products began to increase. Prenatal child insurance is a form of insurance for children that includes a special contract that covers the child during the fetal period. Although NHI coverage starts after the child is born, many Korean parents are interested in the product, since it covers health problems that might have existed before the birth (e.g., congenital malformation), as well as any disease and injuries that could happen to the child after the birth [7]. However, few researchers have studied this phenomenon.

The objective of this study was to investigate pediatric health access by describing the ecology for the medical care of children and adolescents in a medical environment where a well-balanced system between NHI and PMI is required.

## 2. Materials and Methods

### 2.1. Data Source and Study Population

The 2018 Korea Health Panel (KHP) data were used in the present study. The 2018 dataset is the latest publicly released by the National Health Insurance Corporation in collaboration with the Korea Institute for Health and Social Affairs. Detailed information regarding this panel survey is available elsewhere [8]. In 2018, nationally representative panel data for 6379 household, which included 17,008 participants, was collected. This included 2746 individuals aged 18 years old and younger in 2018; these individuals were used as participants of this study.

### 2.2. Variables

#### 2.2.1. Main Variables

Anyone with PMI was defined as “policyholders”, while people without any PMI were defined as “non-policyholder”. According to insurance coverage, PMI status was further divided into three types: fixed amount only (F), indemnity only (I), and mixed (F + I). The number of PMI held, regardless of the PMI type, was categorized into three groups: 1, 2, and ≥3.

The use of medical services were further categorized into ambulatory, inpatient, and emergency care. According to different medical settings (clinics, hospitals, and tertiary hospitals), ambulatory and inpatient care services were further examined.

#### 2.2.2. Covariates

Covariates included age, sex, economic status, and chronic diseases. Age was categorized into four groups: <2 years, 2 to 5 years, 6 to 11 years, and 12 to 18 years [9]. Household income acquired from 1 January to 31 December in the previous year was divided into quintiles based on family size: 1st quintile (poorest)–5th quintile (wealthiest). Any survey participants who reported any doctor-diagnosed diseases were defined as people with chronic disease. Any reported diseases were classified based on the KHP disease codes by interviewers. Medical experts completed additional confirmations regarding the denoted codes. In multivariate analyses, pediatric comorbidity index developed by Sun and colleagues [10] was used for risk adjustment considering the underlying health status of pediatric patients. Since the KHP disease codes provided three-character categories, four-character subcategories could not be specified for F06 (other mental disorders due to brain damage and dysfunction and to physical disease) and F43 (reaction to severe stress, and adjustment disorders), so F06 was encoded as anxiety and F43 as depression in this study.

### 2.3. Statistical Analysis

The number of children and adolescents per 1000 people aged 18 years old and younger participating in health care was estimated to describe the ecology of medical care for children and adolescents. The KHP data collect specific dates of medical use from each survey participant. Based on the reported dates, person–month data were created. When an individual received no medical use in the month, the value “0” was assigned, while the value “1” was assigned if there was one or more medical use in the month. The national estimates were calculated by applying the survey weight and multiplying by 1000. In multivariate analyses, generalized estimating equations were used. An exchangeable structure seemed most appropriate [11], and the models were adjusted for age, sex, economic status, and pediatric comorbidity index. All statistical analyses were performed with SAS version 9.4 (SAS Institute Inc., Cary, NC, USA) and *p*-values of < 0.05 were considered to indicate statistical significance.

### 2.4. Ethical Statement

The ethical evaluation was exempted by the institutional review board of Daegu Catholic University Medical Center (CR-22–007-PRO-001-R).

## 3. Results

Table 1 shows sociodemographic characteristics and PMI enrollment status of study participants. About half of the study participants were aged 12 years old and older, and 24.7% reported doctor-diagnosed chronic diseases. The two most frequently reported diseases were vasomotor and allergic rhinitis (n = 330) and atopic dermatitis (n = 75) (Appendix A). Of the total study participants, 87.3% of children and adolescents had PMI. Among these policyholders, 33.6% had a mixed type, and 44.1% held more than one insurance policy.

Figure 1 illustrates the ecology of medical care for Korean children and adolescents aged 18 years old and younger. Of 1000 children and adolescents, 404 visited a clinic, 67 visited a hospital outpatient department (OPD), 49 visited an OPD in a tertiary hospital, 11 received emergency care, 5 received inpatient care in a hospital, and 9 were hospitalized (5 in a hospital, 3 in a tertiary hospital, and 1 in a clinic) in an average month.

The proportions of children and adolescents receiving medical care in different health setting according to the detailed characteristics of the study participants are shown in Table 2. The use of emergency care was higher in boys than in girls, and the proportions of infants receiving medical services were higher than in older children and adolescents. The study participants from the poorest families received less ambulatory care services than other household income groups, and the use of inpatient care in a tertiary hospital was the lowest in this group.

Table 3 provides the results of the multivariate analyses according to PMI status. After adjusting for sex, age, household income, and pediatric comorbidity index, receiving ambulatory care services in clinics was significantly more likely among children and adolescents with PMI (adjusted odds ratio [aOR] = 1.16 [95% confidence interval [CI]: 1.00–1.35]). Receiving ambulatory care services in clinics was significantly more likely among indemnity type policyholders (aOR = 1.23 [1.05–1.45]), while visiting OPD in hospitals was more likely among policyholders with fixed amount type insurance coverage. Moreover, receiving ambulatory care services in clinics was significantly more likely among single policyholders (aOR = 1.18 [1.00–1.37]), while visiting OPD in hospitals was more likely among multiple policyholders.

## 4. Discussion

The ecology of medical care, a model that demonstrated patient-centered and population-based medical care use, was first described by White and colleagues in 1961 [12]. Although succeeding research was performed worldwide [13,14,15,16,17,18], studies about children and adolescents are limited. A study conducted in the United States in 2003 reported that the ecology of medical care for children and adolescents was similar to that of adults, mostly receiving community-based health care but varied by sociodemographic characteristics, including health insurance status [19]. It was also confirmed that uninsured individuals had lower access to medical services [19]. Compared to the United States, children and adolescents in Japan visited community physicians and hospital-based outpatient clinics more often [20]. Nevertheless, no variables representing medical insurance status were included in the study since Japan has a social insurance system that allows free access to everyone, including children [20]. Unlike the United States and Japan, Korea has a unique medical environment in that it operates a NHI system, which unintentionally allows an active PMI market. As confirmed in the previous study [18], the tendency to receive ambulatory care in tertiary hospitals more than in other countries was also confirmed among Korean children and adolescents. Children and adolescents in Korea showed about 10 times more OPD visits at tertiary hospitals than in Japan [20], and approximately 8 times more OPD visits at hospitals than in the United States [19]. Such high accessibility to hospital-based settings may indicate a lower barrier to the use of medical services among consumers. However, it should not be overlooked that it may indicate a functional weakening of the medical delivery system, as well as excessive use of NHI finances [21,22]. The multivariate analysis of our study, which revealed that multiple policyholders visited OPD in a hospital more than non-policyholders after adjusting associated variables, may indicate that PMI induces further use of medical services in hospital-based settings.

Furthermore, considering the use of medical services among children and adolescents in association with PMI has two additional notable aspects. First, there is an enrollment driving-force issue. Enrollment into PMI for children and adolescents entirely depends on the caregiver’s choice, which is based on the caregiver’s economic status. By examining the PMI enrollment status of 2973 children and adolescents between 2009 and 2012 using the KHP data, those from the poorest families tended to transit to non-policyholders compared to other income groups [23]. Given the recent tacit agreement that PMI enrollment for a child should be provided by one’s parents means that an uninsured fetus could cause a vulnerability in terms of health equity. Moreover, it can solidify the perception that health problems must be solved individually. This makes additional discussion necessary because the situation is outside of the realm of the NHI’s basic concept of pooled financing. Second, PMI could be a prerequisite for easy access to health services, even inducing unnecessary use of health care. PMI products provided in Korea actively compensate for out-of-pocket expenses and are considered a useful supplementary or complementary tool to patients and caregivers that guarantees additional compensation in the NHI system [5]. The only difference between public health insurance and PMI is the timing of the compensation and who makes it, which may not be noticeable to the insured person. There is no management strategy in which PMI compensates for co-payments for covered services, a device created to minimize the development of a moral hazard. The multivariate analysis of this study revealed that PMI policyholders were statistically more likely to use ambulatory services in clinics than non-policyholders, and this phenomenon was clearly confirmed in the indemnity-only type policyholders. Such results may imply an association between active compensation by PMI and the induced use of ambulatory care at clinics. Considering the present study results and the fact that such utilization behavior would continue in adulthood, an appropriate PMI monitoring and management system should be prepared. For ambulatory services in clinics, legal restriction measures for PMI compensation should be established in order to minimize consumers’ moral hazard and provider-induced medical use.

In this study, the proportion of children and adolescents without any PMI was 12.7%, while 19.9% were PMI uninsured in 2009 [23]. Universal health care through insurance enrollment is clearly guaranteed for Korean children and adolescents, a goal that the United States is still trying to achieve. However, enrollment in public health insurance or PMI itself does not guarantee complete comprehensive access to care [24]. Such widespread PMI subscriptions among Korean children and adolescents could be interpreted as consumer dissatisfaction with NHI and explained in association with the constant demand for prenatal child insurance. Insurance coverage not only allows individuals to enter the health delivery system but also determines what kind of services are provided within the system. More specifically, health insurance is one of the factors that determines the specific aspects of the utilization of health services, including type, site, purpose, and time interval [25]. QHPs were introduced as a security device for providing essential health care regarding insurance coverage in the United States [26], but the Korean government may need to consider such benefit standards in terms of the efficient control of supplementary PMI. In addition, the government should review the NHI reimbursement standards for essential health care of children and adolescents, including congenital malformation and child injuries.

The present study has several limitations. Above all, it did not investigate health insurance coverage qualitatively. Other study limitations originate from the KHP data characteristics. First, an additional survey determining symptoms and health problems was only conducted for adults, so the confirmation of clinical symptoms among children and adolescents was not possible. Second, the disease codes provided by the KHP data were not four-character subcategories, so the original SAS code contributed by Sun and colleagues [10] had to be modified. Third, the number of people applying for home health care could not be estimated, since the KHP did not provide such data. Despite these limitations, the significance of this study lies in that it examined the ecology of the medical care of children and adolescents by using nationally representative data in a medical environment where the government-sponsored medical insurance system and a private market coexist. It is expected that the results of this study could be used by many future researchers in other countries interested in achieving bona fide UHC, especially for application and management in PMI market.

## 5. Conclusions

Of paramount importance in achieving UHC in medical care utilization is making efforts to minimize either excess or deficit in healthcare use. Therefore, countries with national health insurance schemes should continuously practice the proper regulation and management of PMI, including reviewing PMI compensation measures, NHI reimbursement standards, and consumers’ perspectives on NHI and PMI. In terms of pediatric health access with regard to formulating the country’s future health status, the optimal level of insurance coverage should be discussed within the political spectrum.

## Figures and Tables

**Figure 1 children-09-01101-f001:**
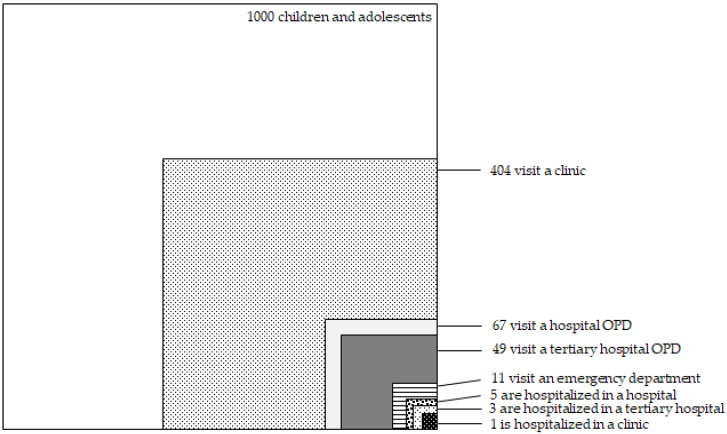
Estimated number of children and adolescents per 1000 people aged 18 years old and younger participating in health care. Abbreviation: OPD—outpatient department.

**Table 1 children-09-01101-t001:** Private medical insurance status and sociodemographic characteristics of study participants.

Variables	Subgroups	Number (%)	*p*-Value
Private medical insurance (PMI)	Non-policyholders	349 (12.7)	<0.0001
	Policyholders	2397 (87.3)	
PMI types (Policyholders only)	Fixed amount type only (F)	483 (20.2)	<0.0001
	Indemnity type only (I)	1109 (46.3)	
	Mixed type (F + I)	805 (33.6)	
Number of PMI policies (Policyholders only)	1	1336 (55.9)	<0.0001
	2	745 (31.1)	
	≥3	313 (13.1)	
Age	<2	82 (3.0)	<0.0001
	2~5	418 (15.2)	
	6~11	854 (31.1)	
	12~18	1392 (50.7)	
Sex	Boys	1388 (50.5)	0.567
	Girls	1358 (49.5)	
Household income	1st (poorest)	96 (3.5)	<0.0001
	2nd	451 (16.4)	
	3rd	778 (28.3)	
	4th	754 (27.5)	
	5th (wealthiest)	667 (24.3)	
Chronic disease ^1^	Yes	677 (24.7)	<0.0001
	No	2069 (75.4)	
Residential area	Urban	1006 (36.6)	<0.0001
	Rural	1740 (63.4)	
Disability	Yes	29 (1.1)	<0.0001
	No	2717 (98.9)	
Total	-	2746 (100.0)	-

^1^ Study participants reported any doctor-diagnosed diseases.

**Table 2 children-09-01101-t002:** Estimated number of children and adolescents per 1000 people aged 18 and younger participating in health care by private medical insurance status and sociodemographic characteristics.

Variables	Subgroups	Ambulatory Care			Inpatient Care			Emergency Care
		Clinics	Hospitals	Tertiary Hospitals	Clinics	Hospitals	Tertiary Hospital	
PMI	Policyholders	402.47 (7.32)	67.73 (4.19)	47.36 (3.23)	1.20 (0.29)	5.16 (0.62)	3.59 (0.46)	11.08 (0.82)
	Non-policyholders	419.50 (19.20)	62.89 (9.97)	60.09 (9.11)	1.18 (0.69)	5.43 (1.57)	4.80 (1.38)	10.93 (2.08)
	*p*-value	0.407	0.655	0.187	0.978	0.874	0.406	0.945
PMI types	F	325.56 (13.60)	61.87 (10.40)	37.43 (5.25)	0.60 (0.31)	3.44 (1.70)	1.78 (0.61)	6.52 (1.31)
	I	465.98 (10.95)	77.15 (6.44)	52.22 (5.27)	1.81 (0.53)	5.72 (0.84)	4.39 (0.78)	13.63 (1.37)
	F + I	346.21 (12.14)	56.00 (5.80)	45.49 (5.00)	0.56 (0.26)	5.29 (1.02)	3.41 (0.70)	9.73 (1.18)
	*p*-value	0.568	0.366	0.457	0.437	0.423	0.217	0.291
Number of PMI policies	1	439.36 (9.91)	70.18 (5.71)	48.47 (4.56)	1.66 (0.45)	5.13 (0.84)	3.88 (0.66)	12.46 (1.19)
	2	357.52 (12.92)	67.76 (7.61)	44.58 (5.05)	0.61 (0.29)	5.70 (1.13)	2.67 (0.58)	7.55 (1.12)
	≥3	322.13 (15.64)	54.93 (9.12)	48.42 (8.07)	0.26 (0.26)	4.00 (1.33)	4.41 (1.45)	12.65 (2.12)
	*p*-value	<0.001	0.240	0.812	0.011	0.737	0.861	0.278
Age	<2	725.59 (26.27)	151.52 (27.02)	72.99 (17.48)	1.04 (1.04)	11.39 (3.36)	7.59 (2.77)	23.90 (5.85)
	2~5	639.23 (15.53)	143.07 (13.46)	62.67 (8.29)	3.40 (1.06)	11.01 (1.84)	5.78 (1.41)	19.33 (2.28)
	6~11	393.35 (10.41)	61.79 (6.05)	53.60 (5.61)	0.67 (0.29)	4.99 (1.10)	3.90 (0.77)	9.76 (1.21)
	12~18	264.78 (6.67)	24.78 (2.38)	36.51 (3.62)	0.50 (0.20)	1.84 (0.42)	2.26 (0.40)	6.66 (0.82)
	*p*-value	<0.001	<0.001	<0.001	0.008	<0.001	0.001	<0.001
Sex	Boys	398.21 (9.42)	70.16 (5.61)	47.60 (3.63)	0.76 (0.24)	5.81 (0.92)	3.76 (0.59)	12.88 (1.08)
	Girls	411.81 (9.92)	63.76 (5.29)	50.65 (4.97)	1.67 (0.48)	4.54 (0.68)	3.75 (0.67)	9.11 (1.09)
	*p*-value	0.320	0.407	0.620	0.093	0.267	0.991	0.013
Household income	1st (poorest)	329.22 (30.82)	49.06 (14.87)	41.35 (12.46)	1.02 (1.02)	7.71 (2.99)	2.66 (1.76)	10.73 (4.10)
	2nd	355.72 (15.22)	80.24 (9.82)	53.60 (7.66)	3.19 (1.32)	5.70 (1.31)	4.56 (1.13)	11.97 (2.21)
	3rd	432.88 (12.82)	81.31 (8.76)	50.48 (5.99)	1.29 (0.43)	6.28 (1.36)	4.55 (0.96)	13.71 (1.59)
	4th	418.19 (12.97)	64.49 (6.33)	51.56 (5.61)	0.48 (0.22)	5.60 (1.07)	3.51 (0.78)	10.27 (1.28)
	5th (wealthiest)	399.16 (14.49)	47.65 (6.73)	42.99 (6.20)	0.62 (0.30)	2.85 (0.80)	2.75 (0.77)	8.35 (1.32)
	*p*-value	0.052	0.006	0.450	0.030	0.022	0.184	0.054
Chronic disease ^1^	Yes	406.27 (13.07)	63.83 (8.23)	81.59 (9.23)	1.01 (0.43)	5.99 (1.58)	4.03 (1.01)	12.38 (1.71)
	No	404.31 (7.98)	68.05 (4.37)	39.28 (2.75)	1.25 (0.32)	4.95 (0.58)	3.67 (0.49)	10.67 (0.85)
	*p*-value	0.898	0.650	<0.001	0.654	0.537	0.753	0.370
Residential area	Urban	400.20 (10.89)	80.63 (7.17)	48.47 (5.00)	0.41 (0.24)	4.19 (0.68)	2.38 (0.58)	9.58 (1.13)
	Rural	407.90 (8.79)	57.76 (4.23)	49.49 (3.83)	1.74 (0.41)	5.89 (0.85)	4.70 (0.63)	12.08 (1.03)
	*p*-value	0.582	0.006	0.871	0.005	0.120	0.007	0.103
Disability	Yes	528.46 (57.97)	161.83 (57.73)	248.85 (60.46)	0.00 (0.00)	23.78 (24.12)	10.28 (6.20)	10.20 (6.27)
	No	403.44 (6.89)	66.06 (3.85)	46.93 (2.98)	1.21 (0.27)	5.00 (0.52)	3.69 (0.44)	11.07 (0.77)
	*p*-value	0.030	0.092	0.001	<0.001	0.428	0.280	0.888

Values are presented as number per 1000 people (standard error). ^1^ Study participants reported any doctor-diagnosed diseases. Abbreviations: PMI—private medical insurance; F—fixed amount only; I—indemnity only; F + I—mixed.

**Table 3 children-09-01101-t003:** Association of private medical insurance and health care participation (reference = non-policyholders).

Variables	Subgroups	Ambulatory Care			Inpatient Care			Emergency Care
		Clinics	Hospitals	Tertiary Hospitals	Clinics	Hospitals	Tertiary Hospital	
PMI	Policyholders	1.16 (1.00–1.35)	1.36 (0.95–1.94)	0.86 (0.59–1.24)	0.83 (0.44–1.56)	1.24 (0.65–2.34)	1.15 (0.32–4.10)	1.22 (0.81–1.82)
PMI types	F	1.04 (0.87–1.25)	1.78 (1.08–2.93)	0.74 (0.47–1.17)	0.48 (0.20–1.17)	1.14 (0.42–3.11)	0.73 (0.15–3.62)	0.84 (0.48–1.45)
	I	1.23 (1.05–1.45)	1.22 (0.84–1.78)	0.88 (0.60–1.31)	0.89 (0.46–1.72)	1.09 (0.55–2.16)	1.33 (0.35–4.99)	1.29 (0.85–1.97)
	F + I	1.09 (0.92–1.30)	1.57 (1.04–2.37)	0.87 (0.57–1.33)	0.91 (0.45–1.84)	1.82 (0.88–3.75)	0.85 (0.19–3.69)	1.27 (0.82–1.99)
Number of PMI policies	1	1.18 (1.00–1.37)	1.19 (0.82–1.73)	0.84 (0.57–1.23)	0.81 (0.42–1.57)	1.04 (0.53–2.03)	1.27 (0.34–4.73)	1.23 (0.81–1.87)
	2	1.14 (0.96–1.36)	1.91 (1.25–2.92)	0.87 (0.56–1.34)	0.72 (0.35–1.48)	1.88 (0.91–3.87)	0.90 (0.21–3.94)	0.98 (0.61–1.57)
	≥3	1.09 (0.90–1.33)	1.87 (1.14–3.06)	0.98 (0.61–1.58)	1.31 (0.54–3.20)	1.69 (0.68–4.20)	0.40 (0.04–3.74)	1.89 (1.14–3.14)

Values are presented as adjusted odds ratios (95% confidence intervals). Adjusted for sex, age, household income, and pediatric comorbidity index. Abbreviations: PMI—private medical insurance; F—fixed amount only; I—indemnity only; F + I—mixed.

## Data Availability

Data supporting reported results can be found at Korea Health Panel Survey (https://www.khp.re.kr) (accessed on 25 October 2021).

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
