# Peer review of "Pediatric Health Access and Private Medical Insurance: Based on the Ecology of Medical Care in Korea"

_children, 2022, doi:10.3390/children9081101_

Round 1
Reviewer 1 Report
The abstract needs to be improved. It must bear reference to the unit of analysis employed in the study.
The conclusion section is also weak and needs to be strengthened. Generally in the conclusion section the following issues should be addressed:
1. Outline the aim of the study
2. A brief reminder of the research methodology and unit of analysis applied in the study.
3. Report on the key findings
4. State the contribution of the study and/ or implications thereof.
5. Identify the limitations of the study.
6. Suggest areas for future research.
Author Response
Thank you for your valuable comments and helpful suggestions.
Please refer to the attached file. Thank you in advance.
Reviewer 2 Report
I found the paper to be generally well written, interesting and informative. I have a couple of suggestions for improvements, in most cases they are of minor significance.
1. on p. 2, starting from line 51 authors write about the private medical insurance in Korea. I'd suggest them to elaborate a bit more about the characteristics/design/statistics of private insurance in Korea.
2. on p. 3, line 94-95 authors write: "Covariates included age, sex, economic status, and chronic diseases. Age was categorized into four groups: <2 years, 2 to 5 years, 6 to 11 years, and 12 to 18 years." It would be good to add justification, why these particular age ranges has been applied.
3. On p. 3, line 126 authors write: "About half of the study participants were aged 12 years old and olde,". This is obvious typo, should be "older".
4. In results section the tables should be edited for better presentation (for example, in table 1 some lines are not aligned)
5. In table 2 it would be good to calculate statistical significance of differences in subsequent groups/subgroups.
6. In the discussion section, p. 6, lines from 194 authors write: "However, it should not be overlooked that it may indicate a functional weakening of the medical delivery system as well as excessive use of NHI finances."
I totally agree, and even suppose this Ismay be the main issue. However, it would be good to find some reference here, so that this is not just speculation.
7. In line authors write: "The multivariate analysis of this study which revealed that multiple policyholders visited 196 OPD in a hospital (...)".
"this", meaning which? Authors' study? If yes, I suggest to write directly „our study”.
8. the statement on p. 7, lines starting from 231, again might be supported with some reference.
9. In general, it seems that expanding the list of references would benefit for the manuscript a lot. Currently it is missing a bit more up-to-date studies, especially in discussion. Some of the references are extremely old. It is not bad to refer to some "canonical" publications, but not necessarily should this be the core of reference list.
10. I’d suggest reformulating the conclusions a bit, so that they are more directly connected with the results of the study, not that generic as they are right now.
Author Response
Thank you for your valuable comments and helpful suggestions. Please refer to the attached file. Thank you.

Reviewer 3 Report
Thanks for the opportunity to review this interesting and significant paper. It's well known that
private medical insurance plays an important role in health access. Also, this paper is easy to read and I believe there will be large audience. I have read the entire manuscript and I have a few comments and suggestions for its authors:
- a summary table with all the variables included in the study.
I particularly appreciate the fact that the authors review the specialized literature in the field, but they used few recent references from top journals. Further on, I suggest the addition of the following references:
· Dragos, S. L., Mare, C., Dragos, C. M., Muresan, G. M., & Purcel, A. A. (2022). Does voluntary health insurance improve health and longevity? Evidence from European OECD countries. The European Journal of Health Economics, 1-15.
· He, G., Li, C., Wang, S., Wang, H., & Ding, J. (2022). Association of insurance status with chronic kidney disease stage at diagnosis in children. Pediatric Nephrology, 1-10.
· Yu, J., Perrin, J. M., Hagerman, T., & Houtrow, A. J. (2022). Underinsurance among children in the United States. Pediatrics, 149(1).
· Zhan, C., Wu, Z., Yang, L., Yu, L., Deng, J., Luk, K., ... & Zhang, L. (2022). Disparities in economic burden for children with leukemia insured by resident basic medical insurance: evidence from real-world data 2015–2019 in Guangdong, China. BMC health services research, 22(1), 1-12.
Still, I think the authors should develop in conclusions future research directions.
Author Response

(The authors gave the same response as above.)
